**Advancing studies on global biocrusts distribution**
Siqing Wang[1,2], Li Ma[1,2], Liping Yang[1,2], Yali Ma[1,2], Yafeng Zhang[3], Changming Zhao[1,2], Ning
Chen[1,2,4*]
[1] State Key Laboratory of Herbage Improvement and Grassland Agro-ecosystems, College of
Ecology, Lanzhou University, Lanzhou, Gansu 730000, China
[2] Yuzhong Mountain Ecosystems Observation and Research Station, Lanzhou University,
Lanzhou, Gansu 730000, China
[3] Shapotou Desert Research and Environment Station, Northwest Institute of Eco-Environment
and Resources, Chinese Academy of Sciences, No.320, Donggang West Road,
Lanzhou, Gansu 730000, China
[4] Instituto Multidisciplinar para el Estudio del Medio "Ramon Margalef", Universidad de
Alicante, Carretera de San Vicente del Raspeig s/n, San Vicente del Raspeig Alicante 03690,
Spain
*Correspondence to Ning Chen, Email: cn@lzu.edu.cn. Tel.: +86-931-8912551. ORCID: 0000-
0002-1779-915X.
**Abstract:** Biological soil crusts (biocrusts hereafter) cover a substantial proportion of dryland
ecosystem and play crucial roles in ecological processes such as biogeochemical cycles, water
distribution and soil erosion. Consequently, studying spatial distribution of biocrusts holds great
significance for drylands, which is still lacking, especially in a global scale. This study aimed
to stimulate global-scale investigations of biocrusts distribution by introducing three major
approaches: spectral characterization indices, dynamic vegetation models, and geospatial
models, while discussing their applicability. Then, we summarized present understandings of
biocrusts distribution. Finally, to further advance this field, we proposed several potential
research topics and aspects, including building standardized database of biocrusts, enhancing
non-vascular vegetation dynamic models, integrating multi-sensor monitoring, making full use
of machine learning, and focusing on regional research co-development. This work is supposed
to significantly contribute to mapping biocrusts distribution, and thereby to advance our
understandings of dryland ecosystem management and restoration.
**Key words:** biological soil crusts; distribution; drylands; global scales; regional scales

**1. Introduction**
Biological soil crusts (biocrusts hereafter) are continuous biotic complexes that live in the

topsoil, which are formed by different proportions of photosynthetic autotrophic (e.g. cyanobacteria, algae, lichens, mosses) and heterotrophic (e.g. bacteria, fungi, archaea) organisms colloidal with soil particles, usually with a thickness of a few millimeters to a few centimeters (Weber et al., 2022). They are able to occupy a wide ecological niche in mid latitudes, polar and alpine regions, covering approximately 11% of the global land area (Porada et al., 2019). In particular, biocrusts can be adapted to water-limited, nutrient-poor and hostile environments, such as arid and semi-arid areas characterized by low ratios of precipitation to potential evaporation (0.05-0.5 mm mm$^{-1}$) (Pravalie, 2016; Read et al., 2014; Weber et al., 2016).

As vital components of dryland ecosystems, biocrusts fulfill many essential ecological functions. They contribute to stabilizing the soil surface and improving soil permeability and water-holding capacity within the upper few centimeters (Sun et al., 2023; Shi et al., 2023; Gao et al., 2017). By participating in a suite of biogeochemical cycles, biocrusts were estimated to contribute to 15% of terrestrial net primary productivity and 40-85% of biological nitrogen fixation (Elbert et al., 2012; Rodriguez-Caballero et al., 2018). They also impact ecohydrological processes by altering soil microclimate and redistributing soil water (Kidron et al., 2022; Tucker et al., 2017). Moreover, biocrusts influence seed capture and soil seed banks (Kropfl et al., 2022), thereby mediating plant growth and community assembly (Havrilla and Barger, 2018; Song et al., 2022). The extent and magnitude of these ecological functions and services depend on the spatial distribution of biocrusts. Therefore, it is crucial to understand their distribution.

Despite the significance of biocrusts, previous studies have primarily focused on their contributions to C and N cycling in varying habitats and climates (Hu et al., 2019; Morillas and Gallardo, 2015), interspecific interactions and biocrusts biodiversity (Machado De Lima et al., 2021; Munoz-Martin et al., 2019), rather than their spatial distribution, particularly at the global scale. Consequently, a systematic and accurate assessment of biocrusts' ecological roles remains challenging. In this study, we firstly sorted out the main research methods for studying biocrusts distribution (section 1), then reviewed the existing knowledge (section 2), and finally proposed strategies to advance the study of large-scale biocrusts distribution (section 3). This work is expected to deepen our understandings of dryland ecosystem processes, and to provide

a scientific basis for dryland ecosystem conservation and their responses to global change.

## 2. Research Methods

In the study of biocrusts distribution, three methods are commonly used: spectral
characterization, vegetation dynamic modeling, and geospatial modeling. This section provides
an overview of these methods, including their basic principles, adaptability, and limitations.

### 2.1 Spectral characterization index

With advances in remote sensing and geo-information technology, spectroscopy provides
a feasible method of characterizing distribution features from a physical point of view.
Differences in absorption or reflection of specific wavelengths of different ground covers can
effectively identify soil surface objects (Rodriguez-Caballero et al., 2015). By identifying
biocrust-specific bands from the reflectance spectral images (Karnieli et al., 1999), it is possible
to construct a presence-absence map of biocrusts distribution (Fig. **1a**).
The crust index (CI) and biological soil crust index (BSCI) are two of spectral
characterization indexes and have been successfully applied in Negev Desert (cyanobacteria-
dominated) (Karnieli, 1997; Noy et al., 2021) and Gurbantunggut Desert (lichen-dominated)
(Chen et al., 2005). For such indicators, it is critical to determine the threshold of spectral bands
that represent biocrusts. For instance, at an aerosol optical depth of 0.2, the BSCI ranges from
4.13 to 6.23, and narrows to 4.58-5.69 with increasingly poor atmospheric conditions. Overly
strict or loose threshold ranges can easily lead to biocrusts omission or misidentification. In
order to improve the accuracy of biocrusts identification, some researchers took advantage of
the hyperspectral sensor's continuous waveband, and created Continuum Removal Crust
Identification Algorithm (CRCIA) (Chamizo et al., 2012b; Weber et al., 2008). Baxter et al.
(2021) innovatively applied the random forest algorithm to spectral feature classification, and
achieved an accuracy of 78.5% in biocrusts recognition. Another two indexes, i.e., the sandy
land ratio crust index (SRCI) and the desert ratio crust index (DRCI), were also introduced by
taking into account the differences between sandy land (vegetation cover FVC <20%) and
desert environments, which could improve the accuracy of the mapping by ~6% (Wang et al.,
2022b).
The spectral characterization method is easy to use, and thus, facilitates the access to
continuous long-term dynamics of biocrusts distribution. However, mosses and vascular plants
are generally mixed up in this case because their reflectance characteristics are close to each
other in all wavelengths especially when mosses are wet, which makes them indistinguishable
(Fang et al., 2015). Therefore, spectral characterization method is mainly applicable to
situations where biocrusts cover is >30% and plants cover is <10% (Beaugendre et al., 2017).
It should be noted that, the existing indexes mostly correspond to biocrusts cover consisting of
specific dominant groups in specific environments, which cannot be directly extrapolated to
areas with high heterogeneous environments (Table 1). Wetting or disturbance may also lead to
large fluctuations in reflectance of different land types and interfere with biocrusts distribution
monitoring (Rodriguez-Caballero et al., 2015; Weber and Hill, 2016).
**2.2 Dynamic global vegetation models (DGVMs)**
Dynamic global vegetation model is another major method to obtain vegetation cover
(Deng et al., 2022). Dynamic global vegetation model mainly focuses on simulating the
biogeochemical processes (e.g. carbon, water cycles), metabolic and hydrological processes of
organisms (Lenton et al., 2016; Porada et al., 2017). The method possesses significant
advantages to map biocrusts distribution because its assumptions have clear biological
implications (Cuddington et al., 2013), yet may lead to poor predictions of global-scale
distributions due to subjective regional experience and insufficient amounts of biocrust data
(Table 1) (Quillet et al., 2010). To use DGVMs, the following procedures need to be taken. The
first step is building model framework, simulating important and interesting processes such as
biocrusts growth and death, nutrient cycle, and water cycle (Jia et al., 2019). So far, there is
only one dynamic global vegetation model targeting at biocrusts - the Lichen and Bryophyte
Model (LiBry) (Fig. **1b**) (Porada et al., 2019; Porada et al., 2013). The second step is
parameterization. Generally, literature and open databases are used to assign physiological
strategies for different types of biocrust, such as photosynthesis, respiration, and nitrogen
emission under the influence of temperature, precipitation, radiation, biological water
saturation, etc. The third step is importing environmental data into the model to obtain the
biocrusts cover at grid points over a specific study region. At last, the results are tuned and
validated against observation data of biocrusts distribution (obtained by literature comparison
and local field observations), and thereby biocrusts distribution map is produced.

**2.3 Geospatial models**

Directly relating vegetation presence or cover to environmental data instead of indirectly via biological processes in DGVMs is another important way to obtain biocrust's distribution (Beaugendre et al., 2017; Fischer and Subbotina, 2014; Skidmore et al., 2011). Classic statistical models can serve for this purpose. However, they still require comprehensive expert knowledge of how environmental factors affect biocrusts (Pearce et al., 2001), which is hard to get and prone to be biased. Geospatial models which integrating machine learning tools with field survey data and remote sensing data is supposed to hold the most promise (Crego et al., 2022). Geospatial model is also known as species distribution model or ecological niche model (Brown and Anderson, 2014; Jiménez-Valverde et al., 2008; Soberon and Nakamura, 2009). The procedures of how to use geospatial modelling are illustrated in Figure 1c(Rodriguez-Caballero et al., 2018): 1) extracting environmental data for the sites where biocrusts observation data are reported, 2) importing the extracted environmental data into the machine learning framework and obtaining the relationship between biocrusts distribution and environmental variables through a specific algorithm (e.g., decision tree algorithms, Bayesian algorithms, artificial neural networks, etc.), 3) simulating biocrusts distribution by extrapolating to the whole study region using the constructed relationships. Nevertheless, geospatial models are black-boxes and largely non-interpretable, and thus, less capable of capturing key mechanisms behind phenomenon, which may limit its applications. One should note that, to avoid confounding model predictions, inclusion of environmental factors should be based on relevance of environmental factors to biocrusts, and still need expert knowledge to a certain degree(Mäkinen et al., 2022). Besides, not only natural conditions such as climate, topography, soil, etc. that affect biocrust distribution, but also data about human activities such as afforestation, trampling, population density, etc. also need to be considered as environmental indicators in the model. Supplied with sufficient computing power and observation data of biocrust distribution, and suitable environmental, geospatial models are supposed to be able to predict biocrust distribution accurately. Therefore, geospatial modelling is considered to be one of the most appropriate methods available (Table 1).

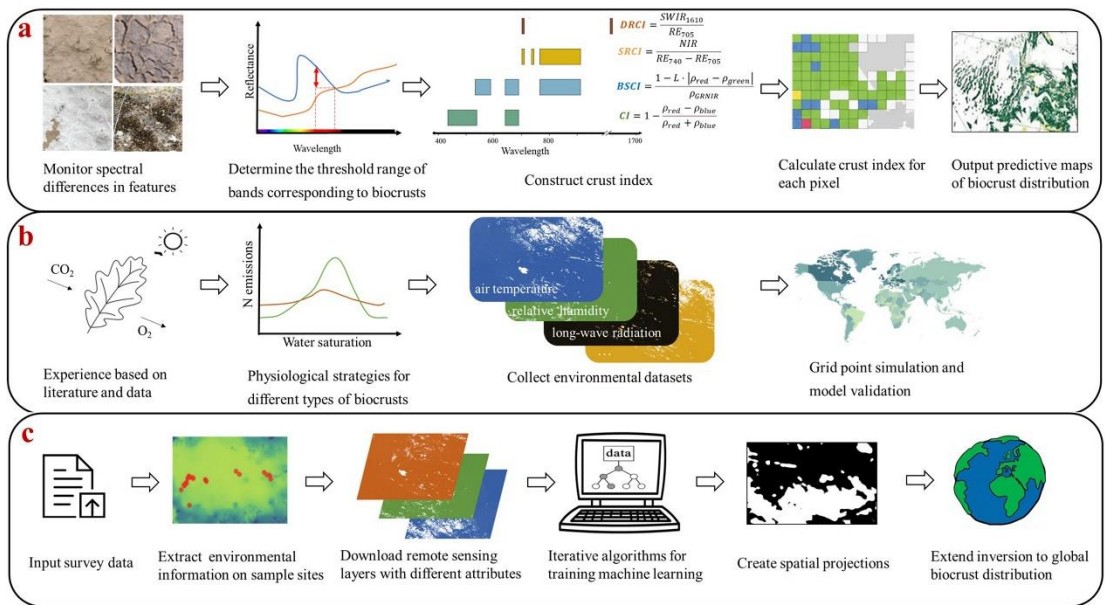


**Fig. 1** Summary of three major approaches of studying biocrust distribution. Illuminations of
applying spectral characterization method **(a)**, vegetation dynamics model **(b)** and geospatial
model **(c)** in biocrusts distribution study. See main text for more detailed introduction to these
methods.
**Table 1** Comparison among the three main types of methods to predict biocrusts distribution

|  | Spectral characteristic index | Vegetation dynamics model | Geospatial model |
|---|---|---|---|
| Principle | Differences in wavelength reflectance of surface features | Differences in the physiological processes of different biocrusts types | Remote sensing information-driven and survey data-based machine learning framework |
| Advantages | Convenience and ease of use | Clear ecological significance | Machine training simulation, without subjective interference |
| Disadvantages | Reflectivity is affected by climate change, disturbances; Mosses and vascular plants have similar reflectance characteristics; | Experience-based promotion with significant human intervention; Experiments need to be supported by big data | Large amount of computing power; Adequate number of sample points to support accuracy |

| | | | |
|---|---|---|---|
| | The results only show the presence or absence of biocrusts, without coverage | | |
| Applicable scales | Regional scale (Desert and sandy land with <20% vegetation cover) | Regional scale Global scale | Regional scale Global scale |


**3.  Current State of Knowledge**

Since 1990, studies on the distribution of biocrusts have been continuously increasing. A group of ecologists, represented by Fernando Maestre (Maestre et al., 2021), David Eldridge(Eldridge and Delgado-Baquerizo, 2019; Eldridge et al., 2023), Matthew Bowker (Qiu et al., 2023), Emilio Rodríguez-Caballero (Rodriguez-Caballero et al., 2018) and others, have actively promoted progress in the field (Fig. 2(a)). The topic has gradually received attention from all over the world, particularly the countries with extensive dryland areas such as China, United States, Spain, United Kingdom, Germany, Australia, and Israel (Fig. 2(b)). However, some other dryland countries and regions, such as central and southern Africa, where biocrust distribution has been reported still there is a paucity of studies and the amount of data on biocrusts is far from adequate (Fig. 2(c)). These areas may be potential areas of widespread distribution of biocrusts in the future.

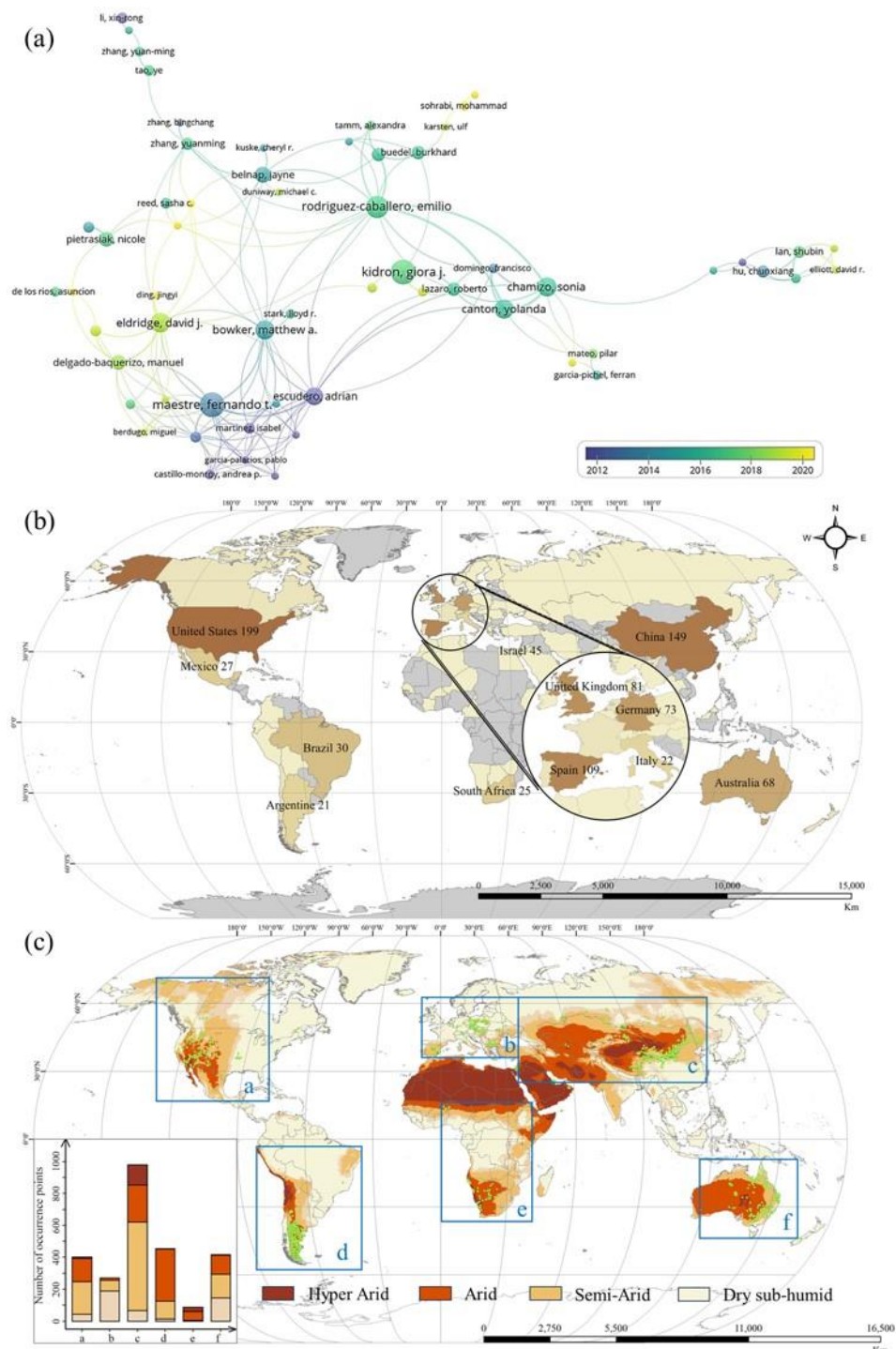

170

Fig. **2** Literature review of biocrust distribution studies. (a) Representative authors associated frameworks for biocrusts distribution studies (1990 to 2022). The time series is the average time of the year of publication, e.g., if the number of articles is 2 in 2004 and 8 in 2019, the node in this figure shows the year as (2004 x 2 + 2019 x 8)/10 = 2016. (b) Map of hotspot countries for biocrust distribution research, with the top 12 countries in terms of number of publications shown; The database is Web of Science, TS = ("biogenic crust*" OR "biological

crust*" OR "biological soil crust*" OR "biocrust*" OR "microphytic crust*" OR "microbiotic

crust*" OR "cyanobacterial*" OR "algal*" OR "lichen*" OR "moss*" OR "biotic crust*") AND

("mapping*" OR "distribution*" OR "spatial pattern*") AND ("dryland" OR "hyper*arid*" OR

"arid*" OR "semi*arid*" OR "dry subhumid*"), with research interests in Environmental

Sciences/Ecology and a total of 700 papers. (c) Global biocrust data distribution, based on field

surveys and literature compilation. Data have been collected and expanded from the published

database (Chen et al., 2020; Rodriguez-Caballero et al., 2018) to 3848 items.

**3.1 Local-regional scales**

At local-regional scales, numerous studies have provided valuable insights into the

distribution patterns of biocrusts in different regions around the world (Fig. 3). In the Mojave

Desert, biocrusts distribution was closely related to geological age, surface stability, topography,

and dust transport (Miller et al., 2004). Lichen, moss, and dark algal crusts patchily distributed

on the desert, averaging 8% cover, though in some bar and shrub zones, the cover could be as

high as 26% (Pietrasiak et al., 2014). In the Colorado Plateau, highly heterogeneous soil matrix

determined the fragmented biocrusts distribution of different types and the wide disparity in

relative abundance and cover (Reynolds et al., 2006; Steven et al., 2013). Collier et al. (2022)

trained drone imagery in the Hawaiian region using timely data collected by cameras and then

successfully mapped watershed-scale biocrusts distribution, predicting cover of ~15-23%. In

the Gurbantunggut Desert, biocrusts cover 28.7% of the area, with a high and uniform biocrusts

cover in the southern part of the desert and a scattered distribution of biocrusts in other areas

(Chen et al., 2005; Zhang et al., 2007). In the Loess Plateau, RGB image-based biocrusts

monitoring showed that variability in biocrusts cover decreased logarithmically with increasing

plot size until a critical size of $1m^2$ after which biocrusts cover remained approximately constant

(Wang et al., 2022a). In Qatar, 26% of the country is covered by biocrusts, with cyanobacterial

biocrusts cover showing a decreasing trend from north-east to south-west (Richer et al., 2012).

In the northern Negev, the distribution of biocrusts is well developed, by simulating the spectral

characteristics of the different components of biocrusts after seasonal precipitation, especially

the chlorophyll absorption characteristics, it was found that the greening of the biocrusts was

obvious, which could provide a basis for the subsequent tracking of their distribution (Panigada

et al., 2019). At the Sinai Peninsula (Egypt) – Negev desert (Israel) border, the distribution dynamics of cyanobacterial biocrusts over a 31-year period has now been obtained from the crust index (Noy et al., 2021). Filamentous cyanobacteria grow in the African Sahara (Issa et al., 1999), where statistical models of combined environmental indicators showed biocrusts cover of 1 ~ 48% and 0 ~ 65% in Banizoumbou and Tamou respectively(Beaugendre et al., 2017).

**3.2 Global scale**

To date, there are only two global-scale studies of biocrusts distribution. Porada et al. (2013) focused on $CO_2$ diffusion rates and photosynthetic processes under dynamic water content saturation in dryland biocrusts. By parameterizing long-term climate and disturbance intervals and averaging simulation results for the past 20 years for each grid point, they estimated that biocrusts cover 11% of global terrestrial land surface (Porada et al., 2019). Their results also showed that the light and dark cyanobacteria were widely distributed in deserts, savannas, grasslands, and Mediterranean woodlands at low latitudes, and increase to some extent with increasing dryness, while mosses were mostly distributed in middle and high latitudes and polar regions (Fig. **3a**). Rodriguez-Caballero et al. (2018) fitted biocrusts presence-only data to the bioclimate, soil properties, land use data, and then extrapolated to a continuous global distribution of biocrusts using the Maxent model. This work assessed the total area covered by biocrusts to be $1.79 \times 10^7$ km$^2$, 12.2% of global drylands, in other words, biocrusts cover was 1.2% larger compared to the area predicted by Porada et al. (2019) (Fig. **3b**). After comparing two maps, biocrust distribution is generally consistent in the large deserts of Asia, western America, Europe and Oceania, while some semi-arid regions, such as the northern and southern margins of African Sahara Desert, South Asia and central North America, have significantly higher biocrusts cover in the latter projection of Rodriguez-Caballero et al. (2018). We estimate that the reason may be that geospatial modeling focuses more on the influence of climate, as the Mediterranean climate and tropical desert climate in the Sahara Desert, as well as the tropical desert climate of northwestern South Asia, which is suitable for biocrust surviving. Additionally, the large number and high cover of biocrust training sets in the central North America could have contributed to the generally high predicted cover in machine learning.

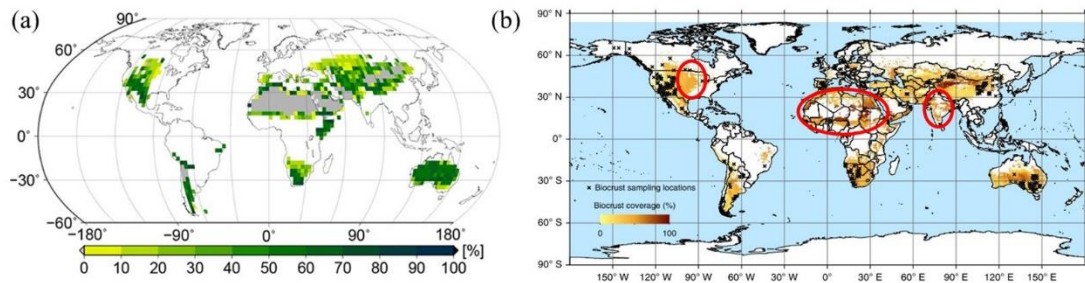

**Fig. 3** Maps of global biocrusts distribution. (a) Prediction based on vegetation dynamic model (Porada et al., 2019). (b) Prediction based on geospatial model (Rodriguez-Caballero et al., 2018).

## 4. Influencing Factors of Biocrust Distribution

Numerous experimental observations and modelling (Kidron and Xiao, 2023; Li et al., 2023; Rodriguez-Caballero et al., 2018) have proved that, on the global scale, biocrust distribution is mainly influenced by water conditions, temperature, soil properties, fire and disturbance (Bowker et al., 2016).

*Water conditions.* In general, total precipitation (Fig. 4(b)) is considered to be critical in determining the distribution of biocrusts (Eldridge and Tozer, 1997). Increased precipitation could lead to higher levels of lichen and moss, while algal cover may first increase and then decrease (Budel et al., 2009; Marsh et al., 2006; Zhao et al., 2014). It should to be noted that, precipitation can also promote the growth of vascular plants, and the continuous and high cover of vascular plants and litterfall will limit the space available to biocrusts (Bowker et al., 2005). In addition to the total amount of precipitation, the seasonality and frequency of precipitation cannot be ignored (Budel et al., 2009). Winter precipitation and/or smaller rain events benefit biocrusts, especially when mean annual precipitation is <500 mm, and high frequency of precipitation can lead to the dominance of biocrusts over vascular plants (Chamizo et al., 2016; Jia et al., 2019). It was experimentally proven that precipitation events of 5 mm were able to maintain normal physiological and ecological functions of the biocrust on the Colorado Plateau, USA, while ever lower precipitation event of 1.2 mm could rapidly kill the moss biocrust (Reed et al., 2012). Non-precipitation water input is another important water resource type. The Namib Desert receives little rainfall, but lichens and moss biocrusts can reach a relatively high cover (~70%) (Budel et al., 2009). This is because local water vapour tends to condense into fog or

dew, which facilitates the survival of three-dimensional species (such as leafy lichens) by trapping air moisture (Eldridge et al., 2020; Kidron, 2019; Li et al., 2021). Similarly, lichen biocrusts are widely distributed in the western U.S. along with the Mexican coast, as the result of the high air humidity (dew formation for almost 1/3 of the year) (Mccune et al., 2022; Miranda‑González and Mccune, 2020).

*Temperature.* Relatively high soil temperature can create an environment of high evaporation that impedes biocrusts colonization (Garcia-Pichel et al., 2013). Regarding air temperature, warming by 4°C could alter biocrust community structure, resulting in a sharp decrease in moss biocrust cover and an increase in cyanobacterial biocrust cover, which became even more significant when warming was interacting with time and precipitation treatments (Ferrenberg et al., 2015). Recent studies have shown that historical and future temperature changes also affect biocrust distribution. For example, the climate legacy over the last 20,000 years could indirectly affect the distribution and relative species richness of biocrusts through changing vegetation cover and soil pH (Eldridge and Delgado-Baquerizo, 2019). Additionally, under future scenarios of increased temperature and aridity, biocrusts cover is predicted to decrease by approximately 25% by the end of the century, with communities shifting towards early cyanobacterial biocrusts (Rodríguez-Caballero et al., 2022).

*Soil properties.* For a long time, it was commonly believed that finer soils benefit biocrusts growth (Belnap et al., 2014; Williams et al., 2013). However, this has been challenged by some scientists (Fig. 4(c)). For example, Kidron (2018) argued that soils with high dust or fine grains were not a necessary condition for biocrusts distribution. Qiu et al. (2023) suggested that soils with small amounts of gravel (0.04-22.34% content, 0.58% best) are more favorable for biocrusts. Another study had shown that the soil parent material determines the degree of surface weathering and the water-holding capacity of the soil, thus indirectly changing the distribution of biocrusts (Bowker and Belnap, 2008). Gypsum or calcareous soils tend to develop mosses and lichens (Elbert et al., 2012), while sandy soils tend to develop cyanobacteria (Root and Mccune, 2012).

*Fire.* Grassland is one major lifeform in dryland ecosystems, making it of great significance to explore the effects of fire events on biocrusts distribution (Palmer et al., 2022).

Fire-induced soil warming can alter the resource allocation and dynamic growth mechanisms
between biocrusts and vascular plants (Mccann et al., 2021), potentially leading to a reduction
in species richness and cover of biocrusts, especially cyanobacteria and algae (Abella et al.,
2020; Palmer et al., 2020). (Condon and Pyke, 2018) showed that moss cover increases with
time after fire, with no significant change in lichen cover.
*Disturbance*. Activities such as grazing, agricultural practices and land development can
significantly impact biocrust distribution. Studies have demonstrated that grazing intensity can
lead to substantial changes in biocrust cover. For instance, in Patagonian rangelands, biocrust
cover decreased by 85%, 89%, and 98% under light, medium, and heavy grazing, respectively
(Velasco Ayuso et al., 2019). In the Loess Plateau, total biocrust cover remained almost
unchanged under light grazing ($< 30.00$ goat dung $/ m^2$), but there were variations in community
structure, with an increase in cyanobacteria biocrusts (23.1%) and a decrease in moss biocrusts
(42.2%) due to the decrease in vascular plant cover (Ma et al., 2023). Tillage practices can
disrupt the soil surface, leading to a reduction in biocrust cover ( 6% on average) and diversity,
and lichens tend to struggle to survive in tilled fields compared to mosses (Durham et al., 2018).
Additionally, late successional biocrusts exhibit higher tolerance compared to pre-successional
biocrusts. Moss biocrusts, for instance, can maintain soil microbial biomass and nematode
abundance better under trampling disturbance compared to cyanobacteria and lichen biocrusts
(Yang et al., 2018). However, contrary to this view, it has been observed that cyanobacterial
biocrusts increased in cover from 81% to 99% after trampling, while lichen and moss biocrusts
decreased from 1.5% and 18% to less than 0.5%. Furthermore, mining activities can
significantly reduce the photosynthetic potential of biocrusts, particularly affecting the recovery
of cyanobacterial biocrusts (Gabay et al., 2022).
*Other factors*. At the global scale, biocrust distribution is also closely linked to
biogeographic isolation. Strong spatial heterogeneity accompanied with spatial distance can set
up barriers to the dispersal of propagules (spores, fungal bodies), which indirectly impedes
colonization of the biocrusts (Garcia-Pichel et al., 2013). In addition, factors such as vascular
plant cover, topography, and solar radiation also influence biocrust distribution, albert to a lesser
extent than the factors mentioned in the above paragraphs. For further insights, readers are

encouraged to consult Chapter 10 of *Biological Soil Crusts: An Organizing Principle in Drylands*, which overview of the control and distribution patterns of biocrust from micro to global scales (Bowker et al., 2016).

To sum, climate is the most important factor of influencing global biocrust distribution, especially in drylands where water is precious to the organisms. But exploration of the roles of climatic factors such as rainfall seasonality and atmospheric drought still needs much more further efforts (Wright and Collins, 2024), especially context of global climate change. Although more attention has been paid to physical properties of soils, the roles of its chemical properties such as the N, P content need to be taken more seriously. Fire and disturbance are usually ignored. Whereas due to the trend towards warmer and drier environments, as well as increasing population and the need to sustain livelihoods, their influences on biocrust distribution may become more important. As one of the basic processes on global scale, biogeographic isolation or changes in land use should be paid more attentions. As amounting data points of biocrust, we can expect this aspect will see a surge in research.

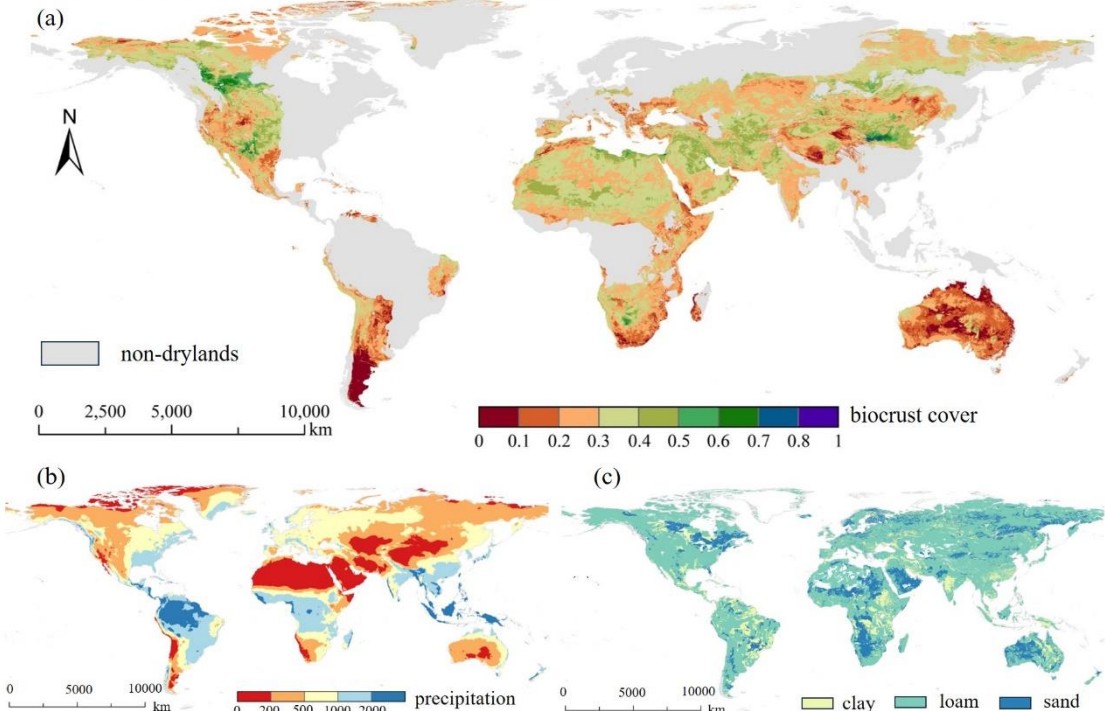

Fig. 4 Biocrust distribution and its critical influencing factors. (a) Biocrust cover map and its influencing factors. (a) Global biocrust distribution, by random forest modelling. Based on a global biocrust database constructed by Chen et al., we expanded the biocrust data to 3848 entries through literature compilation and field surveys and fitted them with four types of

remotely sensed environmental data, including climate, land use, soil properties, and elevation, to finally predict the suitable areas for the biocrust distribution and quantify the biocrust cover. (b) Global average annual precipitation (1970-2020), data from the WorldClim database (version 2.1). (c) Global soil texture distribution, data from HWSD (Harmonized World Soil Database, version 1.2). Precipitation and soil texture were taken as examples of environmental factors.

## 5.  Challenges and Perspectives

Biocrusts are very important for dryland ecosystems, and thus, it is of outstanding significance to understand the current status and dynamics of biocrusts distribution. For the influencing factors (Chapter 4), traditional observation and control experiments provide us with multiple perspectives of basic knowledge. For assessing biocrust distribution patterns (Chapter 3), the methods are shifting from traditional approaches to spectral index, vegetation dynamics and geospatial model, that span multiple subjects like ecology, biology, geology and computer science. However, high-precision biocrusts distribution data across geographic units are still lacking, and research methods are still limited. To further advance studies of biocrusts distribution, we raise the following aspects.

### 5.1 Building standardized biocrusts database

Currently, biocrust data are fragmented, low in volume and accessed from narrow sources, largely limiting spatial prediction from points to areas. Thus, we suggest that a global effort should build a standardized and specialized biocrusts database, consisting of the same data items (main types and cover of biocrusts, latitude, longitude and cover, etc.) and using the same inclusion criteria. It is an important infrastructure for mapping global biocrusts distribution, serving as the benchmark to train and validate spectral characteristics, DGVM, and geospatial models (Engel et al., 2023). Due to the difficulty of conducting field surveys worldwide, compiling biocrusts data from the published literatures or other sources would be the major approach (Fig. 4(a)). To date, several published studies have assembled 900 ~ 1,000 data on biocrust presence or absence from the literature (including 584 data on biocrust cover) (Chen et al., 2020; Eldridge et al., 2020; Havrilla et al., 2019; Rodriguez-Caballero et al., 2018). However, compiling from literatures largely comes to its limitation and is still far from building

a standardized and specialized biocrusts database. While open databases are not specialized to biocrusts, some of them may be important additions (Fig. 5). For example, the biodiversity and specimen datasets such as GBIF and the Atlas of Living Australia (Belbin and Williams, 2015; García-Roselló et al., 2015), which contain a large amount of information on species, of course including mosses and lichens (Table 2), which can contain hundreds or even thousands of entries of biocrusts occurrence or cover. Similarly, global, national and regional plant flora may significantly contribute to building the standardized and specialized biocrusts database. For example, sPlot includes ~2 millions of vegetation plot data (Sabatini et al., 2021), and the European Vegetation Archive (*EVA*) also possesses 1.6 million entries over globe or Europe (Chytrý et al., 2016). Regional datasets like Environmental Monitoring of arid and Semiarid Regions (*MARAS)* surveyed 426 sites (up to September 2020), and provided regular access to 624.50 km$^2$ of rangeland vegetation spatial patterns, species diversity data, soil functional indices, climatic data and landscape photographs in Patagonia region straddling Argentina and Chile (Oliva et al., 2020). Concerns about land use products are also necessary. Global land use maps, based on the PROBA-V sensor, which contain spatial information for Moss & Lichen layer, have an annual update frequency and a resolution of 100 m. In addition to above channels, an increasing number of amateurs are getting involved in science through species identification apps with clean, easy-to-use apps, contributing significantly to global species information entries. The citizen science project *iNaturalist* is a very good example (Wolf et al., 2022). Furthermore, when collecting and collating data from the non-academic sources, the combination of web crawlers and text analysis can help in obtaining biocrusts data and solving key ecological issues.

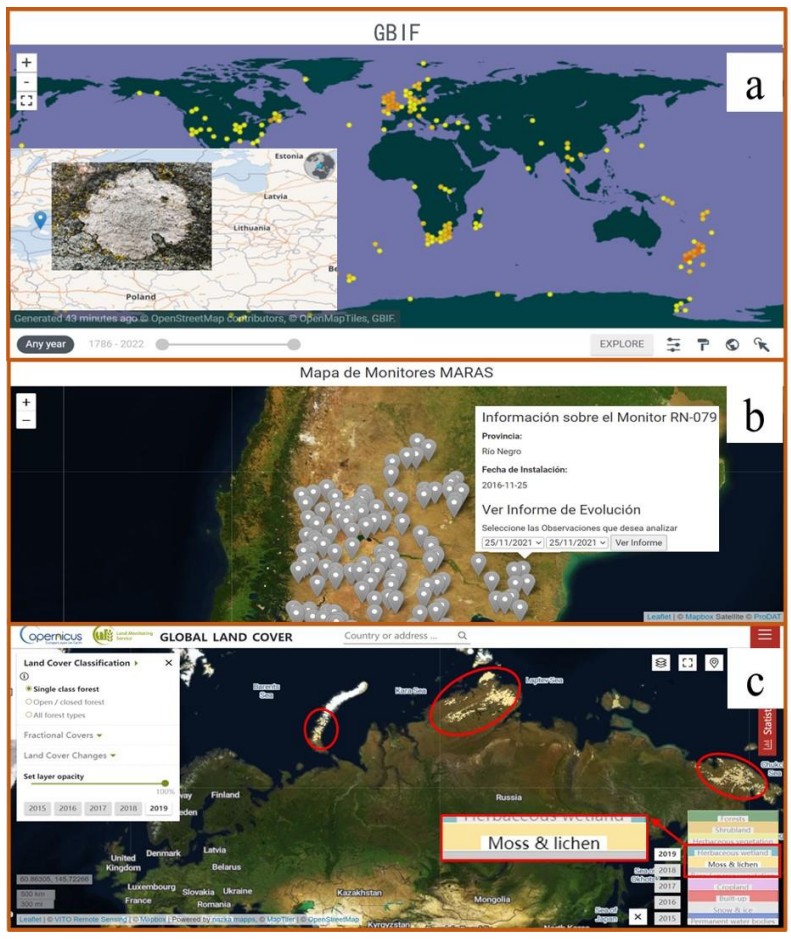

387

388 **Fig. 5** Potential approaches to build a standardized biocrusts database. (a) Distribution of

389 lichens in the GBIF database with an example photo, (b) environmental monitors distribution

390 map of MARAS database, (c) distribution of "mosses and lichens" in the PROBAV_LC100

391 database (light yellow area) in northern Asia, for instance.

392 **Table 2** References for biocrusts database expansion channels

| Data type | Data source | Extend | Biocrust type | Georeferenced records | Presence | Coverage | Link |
|---|---|---|---|---|---|---|---|
| Biodiversity data | the Global Biodiversity Information Facility(GBIF) | Worldwide | Cyanobacteria | ~780000 | ✓ | -- | https://www.gbif.org/ |
| | | | Lichen | ~19000 | | | |
| | | | Moss | ~90000 | | | |
| | Atlas of Living Australia(ALA) | Australia | Cyanobacteria | ~53000 | ✓ | -- | https://www.ala.org.au/ |
| | | | Lichen | ~12000 | | | |
| | | | Moss | ~20000 | | | |
| | Chinese Virtual Herbarium | China | Moss and lichen | -- | ✓ | -- | https://www.cvh.ac.cn/ |
| | Global Plants on JSTOR | Worldwide | Lichen | ~2000 | ✓ | -- | https://plants.jstor.org/ |
| | | | Moss | ~480 | | | |
| Citizen Science | iNaturalist | Worldwide | All | -- | ✓ | -- | https://www.inaturalist.org/ |
| Survey data | MARAS | Argentina and Chile | All | 426 | ✓ | ✓ | https://springernature.figshare.com/collections/The_MARAS_dataset_vegetation_and_soil_characteristics_of_dryland_rangelands_across_Patagonia/4789113 |
| | sPlot | Worldwide | Lichen | 6801 | ✓ | ✓ | https://www.idiv.de/en/splot.html |
| | | | Moss | 11001 | ✓ | ✓ | |
| | GrassPlot | Worldwide | Non-vascular plants | 6623 | ✓ | ✓ | https://edgg.org/databases/GrassPlot/ |
| | Vegbank | Canada and the United States | Moss and lichen | ~15000 | ✓ | ✓ | http://vegbank.org/ |
| | BLM_AIM | The United States | Moss and lichen | 5200 | ✓ | ✓ | https://gbp-blm-egis.hub.arcgis.com/pages/aim |
| | TERN AEKOS | Australia | All | ~300 | | | http://www.aekos.org.au/ |
| Landcover data | PROBAV_LC100 | Worldwide | Moss and lichen | -- | | | https://land.copernicus.eu/global/products/lc |

393

**5.2 Improving non-vascular vegetation dynamic models**

There are only two DGVMs applicable to non-vascular organisms – LiBry and ECHAM6-HAM2-BIOCRUST (Rodriguez-Caballero et al., 2022). Their performances still need to be improved. Considering spatial self-organization of non-vascular organisms (Gassmann et al., 2000), the effects of fire (Thonicke et al., 2001), vegetation-environment feedback processes (Quillet et al., 2010), functional traits (Boulangeat et al., 2012), intraspecific-interspecific interactions (Boulangeat et al., 2014) and seasonal dynamics in current and/or new DGVMs can be future directions. In addition, how the physical properties, photosynthetic capacity, carbon and nitrogen allocation of biocrusts change along environmental gradient are complex and context-dependent, which should be incorporated into DGVMs (Fatichi et al., 2019). Spatial-explicit DGVMs may be one key to effectively improving the accuracy of simulations in future studies, but are data-consuming. Also, biocrusts are significantly influenced by hydrological processes and vice versa (Chen et al., 2018; Whitney et al., 2017), while ecohydrological models based on hydrological processes are rarely connected to global biocrusts distribution predictions. (Jia et al., 2019) tried to incorporate biocrusts cover as a system state variable in an ecohydrological model and investigated biocrusts cover under rainfall gradient. If fed with global data of environmental variables (mainly hydrological relevant ones), ecohydrological models may be a new approach to predicting biocrusts distribution at global scales.

**5.3 Integrated application of high-quality sensors**

Spectral characterization method lies on the differences in spectral reflectance of biocrusts and other land types at different wavelengths, and thus, the accuracy of the results depends on the quality of the sensors. However, previous studies have often used a single sensor with constant band intervals for distinguishing biocrusts, which may result in missed spectral feature identification of land types (Chamizo et al., 2012a). If the biocrusts index can be constructed by combining and comparing the full-band spectral data from multiple terrestrial sensors and infrared cameras and other devices, the errors will be reduced to a certain extent, thus improving the classification accuracy (Wang et al., 2022b). In addition, the unique advantages of hyperspectral data with large data volume and narrow band allow it to be used to combine with observation data to co-develop new biocrusts discrimination standard. If further estimation of

biocrusts cover can be achieved on this basis, it will be an important contribution to the study
of large scale biocrusts distribution (Rodríguez-Caballero et al., 2017). To date, high resolution
sensors have been found to be successful in monitoring lichens and mosses (Blanco-Sacristan
et al., 2021) and the release of such products is something important to look out for in the future.
**5.4 Making full use of machine learning**
Machine learning can be combined with remote sensing products to find complex features
from big data to predict global biocrust distribution (Collier et al., 2022). This data-driven
approach has powerful predictive capabilities, especially for mapping species distribution, and
can largely circumvent the mistakes of missing or misidentifying biocrusts caused by traditional
methods (relying on field measurements to determine threshold ranges) (Wang et al., 2022b).
In the remote sensing image classification problem, mature machine learning algorithms
include support vector machines, single decision trees, random forests, artificial neural
networks, etc. (Yu et al., 2020). Ensemble models combining multiple algorithms have been
widely used in the field of species distribution, but relatively few applications exist in field of
biocrusts prediction. In the future, using machine learning to find parameters for dynamic
models of biocrusts, which may be one of the most promising method to predict biocrusts
distribution (Perry et al., 2022).
**5.5 Regional research synergy development**
Research of biocrust distribution have shown significant spatial and climate imbalances.
Spatially, the study areas that have been conducted are relatively concentrated in countries such
as China, USA, Spain, Australia, and Israel. Although there are large areas of dryland
distributed in Africa (other than South Africa), central Asia, central South America and northern
North America, research on biocrusts in these regions are scarce. Unbalanced regional research
efforts are one of the constraints to advancing studies of global biocrusts distribution. Therefore,
how to coordinate and promote the common progress of regional research is an urgent issue at
present. Climatically, in addition to the drylands, the cold zones may be another important area
to explore biocrusts distribution (Pushkareva et al., 2016). On the Tibetan Plateau, studies have
investigated the spatial variation of different types of biocrusts communities across climatic
gradients and their effects on soil temperature features and freezing duration (Ming et al., 2022;
Wei et al., 2022). This urges the studies of biocrusts distribution in the alpine areas.
**6. Conclusion**
This work aims to advance global knowledge of biocrust distribution for better ecosystem
management and sustainable development in drylands. We firstly compared the advantages,
disadvantages, and applicability among three methods, spectral characterization index,
dynamic global vegetation models and geospatial models, in order to provide the most
appropriate methodological suggestions for biocrust distribution studies at different scales and
needs. Then, we systematically sorted out the regional-global biocrust distribution cases, and
drew a map of global biocrust distribution hotspots and a map of spatial distribution of data
points. Further, we tried to clarify the causes of biocrust distribution from several aspects, such
as precipitation, temperature, soil, fire, and other anthropogenic factors. Finally, from a personal
point of view, we would like to focus more on the following points in the future: database
construction, model performance enhancement, big data processing, and synergistic progress
of potential distribution area studies.
**Acknowledgements**
We are grateful to Zhiguang Zhao, Lixun Zhang, Chao Guan for helpful discussions when
preparing the first draft. The study was supported by the National Natural Science Foundation
of China (grant number 32271620, 31971452), National Key Research and Development
Program of China (2023YFF0805603), and the Central University Basic Research Funds
(lzujbky-2021-ey16, lzujbky-2023-eyt01).

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
