# Peer review of "Running title: Global biocrusts distribution"

_EGUsphere, 2023_

## Author Comment (AC1)

**Referee 1:**

*This MS provides a literature review on the mapping of biological soil crusts and presents the author's perspectives on future developments. The study of biological soil crust mapping is an intriguing and essential research direction. While there are existing studies on mapping, a comprehensive synthesis of these efforts is lacking, making the author's research significant. Although the descriptive aspects of the MS are well-done, it lacks theoretical depth and the author's viewpoints. Therefore, major revisions are necessary before considering publication. Specific issues include:*

**Response: Thank you for acknowledging the merits of this study. We are so** happy to contribute to advance studies on mapping biocrust distribution. In the revised manuscript, we have revised the manuscript according to your comments and suggestions. We hope that you will agree with us.

*In section 2.2, the limitations of this model should be addressed, particularly regarding monitoring areas, as no model can feasibly cover anywhere.*

**Response: We totally agree with you. In the revised manuscript, we have acknowledged the disadvantages of dynamic modelling of vegetation and related them to Table 1** "The method possesses significant advantages to map biocrusts distribution because its assumptions have clear biological implications (Cuddington et al., 2013), yet may lead to poor predictions of global-scale distributions due to subjective regional experience and insufficient amounts of biocrust data (Table 1) (Quillet et al., 2010)." (lines 107-111). To address this bottleneck, we provided suggestions in subsections 2 and 4 of the chapter "Challenges and Perspectives", such as integrating vegetation with more comprehensive biogeochemical and hydrological processes and processing big data based on machine learning for modelling the predicted distributions. (lines 395-412, 429-440)

*In section 2.3, the influence of human activities on the growth of biological crusts in many areas, such as afforestation, should be considered. This may not necessarily be related to local climatic conditions and should be included in section 2.3.*

**Response: Thank the reviewer for the suggestion.** In the revised manuscript, we have added a note to the environmental information referred to in the geospatial model "Besides, not only natural conditions such as climate, topography, soil, etc. that affect biocrust distribution, but also data about human activities such as afforestation, trampling, population density, etc. also need to be considered as environmental indicators in the model." (lines 144-147)

*In section 3.1, it is recommended to incorporate research hotspot maps or tables, categorizing cited and uncited literature and research areas. This provides readers with an intuitive understanding of the research hotspots and identifies regions where research has not yet been initiated.*

**Response: The reviewer gets a point. In the revised manuscript, we have updated the new Figure 2** with maps of research hotspot countries and location maps of global biocrust distribution based on field surveys and literature compilation.

[Figure]

Fig. 2 Literature review of biocrust distribution studies. (a) Representative authors associated frameworks for biocrusts distribution studies (1990 to 2022). The time series is the average time of the year of publication, e.g., if the number of articles is 2 in 2004 and 8 in 2019, the node in this figure shows the year as (2004 x 2 + 2019 x 8)/10 = 2016. (b) Map of hotspot countries for biocrust distribution research, with the top 12 countries in terms of number of publications shown; The database is Web of Science, TS = ("biogenic crust*" OR "biological crust*" OR "biological soil crust*" OR "biocrust*" OR "microphytic crust*" OR "microbiotic crust*" OR "cyanobacterial*" OR "algal*" OR "lichen*" OR "moss*" OR "biotic crust*") AND ("mapping*" OR "distribution*" OR "spatial pattern*") AND ("dryland" OR "hyper*arid*" OR "arid*" OR "semi*arid*" OR "dry

subhumid*"), with research interests in Environmental Sciences/Ecology and a total of 700 papers. (c) Global biocrust data distribution, based on field surveys and literature compilation. Data have been collected and expanded from the published database (Chen et al., 2020; Rodriguez-Caballero et al., 2018) to 3848 items.

*Lines 218-222: Inconsistencies can be circled in the fig and analyzed to determine why they are higher in Rodriguez Caballero et al. (2018).*

**Response: In the revised manuscript, inconsistencies have been marked with circles** and explained accordingly in the text. "We estimate that the reason may be that geospatial modeling focuses more on the influence of climate, as the Mediterranean climate and tropical desert climate in the Sahara Desert, as well as the tropical desert climate of northwestern South Asia, which is suitable for biocrust surviving. Additionally, the large number and high cover of biocrust training sets in the central North America could have contributed to the generally high predicted cover in machine learning." (lines 229-234)

*Section 3.3, the first question is why these influencing factors should be selected. The second issue is that the theoretical expression lacks depth, and some visual or intuitive numbers should be given. The entire 3.3 section only has two numbers, which is regrettable. The third issue is that this paragraph lacks a summary and the author's viewpoint, not just descriptive language. The fourth question, the author uses two paragraphs to describe. The first paragraph describes the impact of climate on biocrust, while the second paragraph mainly describes the impact of soil on biocrust. Additionally, the author adds a sentence about the impact of human activities. Is this structure reasonable.*

**Response:Thank you for giving these wonderful suggestions about organization of this section. In the revised manuscript, the following improvements have been made to the original section 3.3:**

**Q1,** In the revised manuscript, we have clarified that these factors are selected according to previous understandings that are mainly based on previous empirical and modelling studies "Numerous experimental observations and modelling (Kidron and Xiao, 2023; Li et al., 2023; Rodriguez-Caballero et al., 2018) have proved that, on the global scale, biocrust distribution is mainly influenced by water conditions, temperature, soil properties, fire and disturbance (Bowker et al., 2016)." (lines 240-243). Specifically, according to Bowker's book chapter (10), the dominant factors of biocrust distribution are variable at different spatial scales. In the article, we mainly chose the most relevant factors for biocrust distribution at the global scale which has been clarified in the article.

**Q2,** based on the theoretical description, we added some literature as an extension, as well as using figures as much as possible to present the ideas quantitatively. (lines 265-276, 294-311)

**Q3 and Q4,** as suggested by the reviewers, we have restructured the chapter structure, turning section 3.3 into a stand-alone chapter 4 "Influencing factors of biocrust distribution". In addition, each of water conditions, temperature, soil properties, fire, disturbance and other factors were discussed in a separate paragraph. At the end of Chapter 4, we have added a summary paragraph to show our points about this issue, "To sum, climate is the most important factor of influencing global biocrust distribution, especially in drylands where water is precious to the organisms. But exploration of the roles of climatic factors such as rainfall seasonality and atmospheric drought still

needs much more further efforts (Wright and Collins, 2024), especially context of global climate change. Although more attention has been paid to physical properties of soils, the roles of its chemical properties such as the N, P content need to be taken more seriously. Fire and disturbance are usually ignored. Whereas due to the trend towards warmer and drier environments, as well as increasing population and the need to sustain livelihoods, their influences on biocrust distribution may become more important. As one of the basic processes on global scale, biogeographic isolation or changes in land use should be paid more attentions. As amounting data points of biocrust, we can expect this aspect will see a surge in research". (lines 321-331)

*Line 248: Confirm if it should be "20,000 years."*
**Response: We've checked and confirmed it's 20,000 years.**

*Lines 312-315: Clarify the relationship between high-resolution imagery and the database.*
**Response: Thank you for pointing out the issue. In the revised manuscript,** we have moved this sentence to the end of the paragraph in section 5.3, "Integrated application of high-quality sensors". (lines 425-427)

*Lines 347-349: Provide examples or precedents for this point. If none exist, explain the scientific basis for this method.*
**Response: In the revised manuscript,** we have cited a reference (Wang et al., 2022) to show that a case that it used multiple sensors to construct biocrust indices, improving the accuracy of cover prediction by 6% ("If the biocrusts index can be constructed by combining and comparing the full-band spectral data from multiple terrestrial sensors and infrared cameras and other devices, the errors will be reduced to a certain extent, thus improving the classification accuracy (Wang et al., 2022b)."; line 418-421).

*The conclusion lacks an overall summary of the entire article. The author is encouraged to provide a concluding paragraph that synthesizes the key findings and insights.*
**Response: Thank you for the suggestion. In the revised manuscript, a new chapter 6 on conclusion was written** "This work aims to advance global knowledge of biocrust distribution for better ecosystem management and sustainable development in drylands. We firstly compared the advantages, disadvantages, and applicability among three methods, spectral characterization index, dynamic global vegetation models and geospatial models, in order to provide the most appropriate methodological suggestions for biocrust distribution studies at different scales and needs. Then, we systematically sorted out the regional-global biocrust distribution cases, and drew a map of global biocrust distribution hotspots and a map of spatial distribution of data points. Further, we tried to clarify the causes of biocrust distribution from several aspects, such as precipitation, temperature, soil, fire, and other anthropogenic factors. Finally, from a personal point of view, we would like to focus more on the following points in the future: database construction, model performance enhancement, big data processing, and synergistic progress of potential distribution area studies." (lines 455-466)

*Additionally, review the entire manuscript for grammar, capitalization, and singular/plural form correctness.*

**Response: In the revised manuscript, we have carefully checked the grammar and writing for several times. Furthermore, Large Language Model – ChatGPT-3.5 was employed to check grammar and to avoid typos.**

---

## Author Comment (AC2)

**Referee 2:**

This manuscript provides a summary of existing knowledge about biocrust distribution, identified factors that relate to the distribution, gaps in global distribution knowledge, and proposed tools to expand the knowledge. I proposed that potential reorganization of the paper and attention to topic sentences and grouping similar information together will help the reader efficiently find the most useful components of the paper.

**Response: Many thanks for your efforts to improve this manuscript, Eva.** In the revised manuscript, we have reorganized the main text to give a clearer logic for readers. For example, the original section 3.3 has been reframed as the new section 4, to explicitly introduce influencing factors of biocrust. The updated framework of main text includes Research Methods (section 2) - introduce how to study this topic, Current State of Knowledge (section 3) - what do we know about biocrust distribution, Influencing Factors of Biocrust Distribution (section 4), Challenges and Perspectives (section 5) – how to advance this topic. We believe that the reorganized framework is much clearer and smoother for readers to follow up.

*L34. suggest update to "continuous biotic complexes" to preface the next part of the sentence*
**Response: Done.**

*L35. This seems to narrow the focus from all photosynthetic organisms that live at the soil surface (e.g. mosses found when glaciers retreat that are early successional stages) to just arid and semiarid. Later (L 212) they do talk about middle latitudes and polar regions.*
**Response: Thank you for the suggestion. In the revised manuscript, we have replaced this expression with** "They are able to occupy a wide ecological niche in mid latitudes, polar and alpine regions, covering approximately 11% of the global land area (Porada et al., 2019). In particular, biocrusts can be adapted to water-limited, nutrient-poor and hostile environments, such as arid and semi-arid areas characterized by low ratios of precipitation to potential evaporation (0.05-0.5 mm mm-1) (Pravalie, 2016; Read et al., 2014; Weber et al., 2016).". (lines 38-42)

*L45. Elbert is a modeled amount rather than an observed amount, I suggest revising to "biocrusts were estimated to contribute 15%..."*
**Response: Done. Please find the revised lines** 46-48 "By participating in a suite of biogeochemical cycles, biocrusts were estimated to contribute to 15% of terrestrial net primary productivity and 40-85% of biological nitrogen fixation (Elbert et al., 2012; Rodriguez-Caballero et al., 2018).".

*L53. What is a carbon and nitrogen mechanism?*
**Response: Thanks for pointing out the confusing point. In the revised manuscript, the sentence has been rewritten** "Despite the significance of biocrusts, previous studies have primarily focused on their contributions to C and N cycling in varying habitats and climates (Hu et al., 2019; Morillas and Gallardo, 2015), interspecific interactions and biocrusts biodiversity (Machado De Lima et al., 2021; Munoz-Martin et al., 2019), rather than their spatial distribution, particularly at the global scale." (lines 55-59)

*L68. Please revise the topic sentence to clarify the scope of the paragraph for the reader.*

**Response: In the revised manuscript, we have rewritten the first sentence to lead the paragraph,** "With advances in remote sensing and geo-information technology, spectroscopy provides a feasible method of characterizing distribution features from a physical point of view." (lines 70-71)

*L104. Revise to "focus"*
**Response: Done.**

*L185. The shift from statements to question was a little bit confusing – should this be a separate paragraph and leave the rest of the paragraph to summarizing the information from these different countries?*
**Response: We have replaced it with a declarative sentence in the revised manuscript, "**In the Loess Plateau, RGB image-based biocrusts monitoring showed that variability in biocrusts cover decreased logarithmically with increasing plot size until a critical size of 1m2 after which biocrusts cover remained approximately constant (Wang et al., 2022a).**"** (lines 197-200)**.** As the context is juxtaposed to this sentence, showing separate findings from around the globe, the paragraphs no longer need to be separated. Thanks for the suggestion.

*L200. These sentence likewise feels like a different topic – the previous part of that paragraph are defining the snapshots of distribution information and then this sentences talks about changes with future scenarios; may be worth moving it into in a separate paragraph, perhaps after the paragraph about factors that influence biocrust distribution (228).*
**Response: We agree with you.** In the revised manuscript, we have moved this part to Chapter 4, Influencing factors of biocrusts distribution - temperature paragraph. (lines 273-276)

*L227. I generally wonder if the order of this manuscript can be revised for clarity – on line 278, the authors say that traditionally, biocrust distribution methods were based on observational and controlled experiments, so this summary of factors that determine distribution – were these based on those three methods (spectral, vegetation dynamics, geospatial)? Or on the observational/experimental? If so, it may make more sense to summarize what is already known from traditional methods before discussing what new information can be gathered from these remote sensing-assisted options. If the structure is not changed, the authors should however be really clear about methods that were used in different parts of the paper so the reader can clearly discern what is the gap in knowledge so that the author's proposed next steps are in context of the broader field.*
**Response: The points you raise are great. In the revised manuscript, we have systematically pointed out the contribution of different research methods in helping to solve the biocrust distribution issue** "For assessing biocrust distribution patterns (Chapter 3), the methods are shifting from traditional approaches to spectral index, vegetation dynamics and geospatial model, that span multiple subjects like ecology, biology, geology and computer science." (lines 346-349)

*L228. I suggest being consistent – either have questions as section headers throughout, or remove the occasional instances.*
**Response: In the revised manuscript, we have replaced the occasional interrogative sentences with narrative sentences in section titles to make the text clearer** (lines 244-320).

*L228. I assume this refers to total precipitation because later the authors discuss seasonality/frequency. It will help the reader if the metric is described explicitly and specifically up front.*

**Response: Yes, precipitation means total precipitation here.** In the revised manuscript, we had replaced "precipitation" with "total precipitation" and rephrased the sentence as "In general, total precipitation (Fig. 4b) is considered to be critical in determining the distribution of biocrusts (Eldridge and Tozer, 1997)." (lines 244-245).

*L235. In what situations did small rain events benefit biocrusts most? That will set up the reader for the contrast with the moss die-off in the Colorado Plateau.*

**Response: In the revised manuscript, we have added the supplementary explanation of this issue** "Winter precipitation and/or smaller rain events benefit biocrusts, especially when mean annual precipitation is <500 mm, and high frequency of precipitation can lead to the dominance of biocrusts over vascular plants (Chamizo et al., 2016; Jia et al., 2019). It was experimentally proven that precipitation events of 5 mm were able to maintain normal physiological and ecological functions of the biocrust on the Colorado Plateau, USA, while ever lower precipitation event of 1.2 mm could rapidly kill the moss biocrust (Reed et al., 2012)." (lines 251-257).

*L240. I think that fog has also been ascribed to biocrust in the Columbia Basin, WA/OR USA. Check papers from the McCune Lab.*

**Response: Thank you for letting know the works of McCune's lab,** and we've cited McCune's papers on the subject in the revised manuscript "Additionally, under future scenarios of increased temperature and aridity, biocrusts cover is predicted to decrease by approximately 25% by the end of the century, with communities shifting towards early cyanobacterial biocrusts (Rodríguez-Caballero et al., 2022)." (lines 273-276).

*L243. This transition to temperature is confusing because no where else in the paragraph was temperature mentioned. Please make a separate paragraph and flesh out the impacts of temperature and the interaction with moisture.*

**Response:In the revised version, we have moved the contents on temperature as a separate paragraph to distinguish the effects of water conditions on biocrust distribution** (lines 265-276).

*L246. What is the relevant information for the reader from the Ferrenberg study that relates to the rest of this paragraph?*

**Response: In the revised manuscript, we have re-summarized the main findings of Ferrenberg's study** "Regarding air temperature, warming by 4°C could alter biocrust community structure, resulting in a sharp decrease in moss biocrust cover and an increase in cyanobacterial biocrust cover, which became even more significant when warming was interacting with time and precipitation treatments (Ferrenberg et al., 2015)." (lines 266-270)

*L247. Again, this sentence would better serve the reader if the "state of knowledge" section about precipitation patterns and biocrust distribution were separate from "how climate has or will change*

*and effects on biocrusts" paragraph or paragraphs.*

*L262. This discussion of disturbance should be a separate paragraph from the one discussing soil parent material and characteristics to help the reader clearly follow the key information.*

*L270. This sentence doesn't make sense next to the intensification of human disturbance topic. Also, please summarize the key information from the Bowker 2016 publication for the reader.*

**Response: Thank you. We agree with your suggestions. In the revised manuscript**, we have transformed the initial section 3.3 into chapter 4, "Influencing Factors of Biocrust Distribution", while describing each of the influencing factors (water conditions, temperature, soil properties, fire and disturbance) of biocrust distribution in global drylands as a separate paragraph (lines 244-311). Finally, for the content of Bowker's book, we made additional notes " For further insights, readers are encouraged to consult Chapter 10 of Biological Soil Crusts: An Organizing Principle in Drylands, which overview of the control and distribution patterns of biocrust from micro to global scales (Bowker et al., 2016)." (lines 317-320).

*L277. Please revise this sentence for clarity: "… and thus biocrust distribution gradually becomes a hot spot since the turn of the century" – is this "hot spot" referring to the importance for dryland ecosystems? Is it referring to the number of publications?*

**Response:** In the revised manuscript we have reorganised the sentence as "Biocrusts are very important for dryland ecosystems, and thus, it is of outstanding significance to understand the current status and dynamics of biocrusts distribution." (lines 343-344)

*L284. Revise to include verb. "We suggest that a global effort should build a standardized and specialized… " or something like that.*

**Response: Done as suggested** (the revised lines 354-357)**.**

*L288. I think that this paragraph is trying to set up the difference between traditional methods which are compiling available information from the literature with building a database with instructions for new observations that would ensure that the same data items and inclusion criteria are added. Adding a topic sentence that sets out explicitly the purpose of this paragraph rather than initially discussing the recommendation may help clarify.*

**Response: As you suggested, in the revised version, the topic sentence has been added at the beginning of the paragraph to clarify the theme** "Currently, biocrust data are fragmented, low in volume and accessed from narrow sources, largely limiting spatial prediction from points to areas." (lines 353-354).

*L319. PROVAV_LC100 database not discussed in text while other databases in that figure were discussed in text.*

**Response: Thank you for the reminder. A description of the PROVA_LC100 data has been added to the text.** "Concerns about land use products are also necessary. Global land use maps, based on the PROBA-V sensor, which contain spatial information for Moss & Lichen layer, have an annual update frequency and a resolution of 100 m." (lines 378-380)

*L369. Revise topic sentence to include climatic characteristics in addition to spatial characteristics for this paragraph.*

**Response:In the revised manuscript, we rephrased the topic sentence as** "Research of biocrust distribution have shown significant spatial and climate imbalances." (line 442).

*L380. Please revise – biocrusts are not an organizing principle, they were previously described as continuous complexes.*

Response: **In the revised manuscript, we removed this sentence** and added a new conclusion paragraph to organize the purpose and results of this work "This work aims to advance global knowledge of biocrust distribution for better ecosystem management and sustainable development in drylands. We firstly compared the advantages, disadvantages, and applicability among three methods, spectral characterization index, dynamic global vegetation models and geospatial models, in order to provide the most appropriate methodological suggestions for biocrust distribution studies at different scales and needs. Then, we systematically sorted out the regional-global biocrust distribution cases, and drew a map of global biocrust distribution hotspots and a map of spatial distribution of data points. Further, we tried to clarify the causes of biocrust distribution from several aspects, such as precipitation, temperature, soil, fire, and other anthropogenic factors. Finally, from a personal point of view, we would like to focus more on the following points in the future: database construction, model performance enhancement, big data processing, and synergistic progress of potential distribution area studies." (lines 455-466).

*L382. Suggest remove the study "this study summarized" sentence unless the authors next these conclusions into the broader literature.*
**Response: Done.**

*Thank you for your work and I hope my review is helpful to maximize the impact of your research.*
**Response: Thank a lot for your patience and detailed comments, Eva. We hope that with revisions the work will be even better!**

---

## Author Comment (AC3)

**Referee 3:**

*General Comments*

*This MS presents a review of the current knowledge of biocrust distribution, and although the authors have cited the appropriate papers and have accurately identified the current state of the research, the MS requires reorganization and a cohesive synthesis of the current state of knowledge. I have addressed potential ways of doing this in the specific comments below. Also, as the third referee, I read the responses of the others and agreed with their comments. I hope that my comments, in addition to theirs, are helpful during the revision of this MS.*

**Response: Thank you for acknowledging the potential and give the helpful feedback on this work. In the revised manuscript,** we have reorganized the manuscript to give a clearer picture of the underlying knowledge framework. As responded above, we have adjusted the section 3.3 to a separate section, i.e., new section 4, to introduce influencing factors of biocrust, which was previously integrated into the section 3 where was used to introduce current knowledge of biocrust distribution. Meanwhile, the new section 4 had been totally rewritten to make it clearer. Furthermore, a new conclusions section, i.e., section 6, was added in the revised manuscript to summarize the main ideas given by this work, to make it smoother for readers to follow up. In addition, we have added a few figures (Fig. 2 and 4) to quantitatively summarize current understanding of biocrust distribution.

*Specific Comments*

*Section 3, titled "what have we known", should be rephrased to a statement rather than a question. I suggest "Current State of Knowledge" or something similar.*

**Response: Thank you for giving this suggestion.** In the revised manuscript, we replaced the title with "Current State of Knowledge" as suggested (line 158).

*In the first paragraph of section 3, citations should reference the work of the authors you mentioned so the reader can find previous studies on biocrust distribution. In this section, it would also be helpful to have a map of all the places where biocrust distribution has been measured. Most studies provide lat and long values, which can easily be mapped and is a good way to show the gaps in our knowledge. It should also be mentioned that although biocrust distribution has been heavily focused on arid and semi-arid regions, several arid and semi-arid regions have received very little attention (especially the Global South). With a map, this can easily be explained in the text.*

*Additionally, Figure 2 (the author framework) does not provide much new scientific information other than demonstrating which authors are the primary experts in the field. It would be more informational to show a schematic with hubs at the most researched locations (I expect China and Utah would dominate), again this would highlight the gaps in the research that need to be filled.*

*In section 3.1, this information may also be displayed on the map I mentioned previously, or in a table. It is difficult, as the reader, to understand how all the values relate to each other and to place them in a geographic context. Also, within the table, it would be good to specify the scale of the data for each study to provide greater context when comparing it against other studies.*

**Response: We are grateful to the reviewer for these valuable suggestions, which are very important to improve this manuscript.** In the revised manuscript, we mainly made the following

modifications:

1) In the first paragraph of section 3, the references of several authors mentioned in this paragraph have been cited in text (lines 159-163);

2) Fig. 2 has been updated by adding a figure of co-occurrence network of researchers (Fig. 2(a)), maps of research hotspot countries (Fig. 2(b)) and biocrust distribution data under different drought gradients (Fig. 2(c)), so as to clearly show readers the most important information from a spatial perspective;

3) In the revised manuscript, we added interpretation of Figure 2(b) and (c) "The topic has gradually received attention from all over the world, particularly the countries with extensive dryland areas such as China, United States, Spain, United Kingdom, Germany, Australia, and Israel (Fig. 2(b)). However, some other dryland countries and regions, such as central and southern Africa, where biocrust distribution has been reported still there is a paucity of studies and the amount of data on biocrusts is far from adequate (Fig. 2(c)). These areas may be potential areas of widespread distribution of biocrusts in the future." (lines 163-169)

[Figure]

Fig. 2 Literature review of biocrust distribution studies. (a) Representative authors associated frameworks for biocrusts distribution studies (1990 to 2022). The time series is the average time of the year of publication, e.g., if the number of articles is 2 in 2004 and 8 in 2019, the node in this figure shows the year as (2004 x 2 + 2019 x 8)/10 = 2016. (b) Map of hotspot countries for biocrust distribution research, with the top 12 countries in terms of number of publications shown; The database is Web of Science, TS = ("biogenic crust*" OR "biological crust*" OR "biological soil crust*" OR "biocrust*" OR "microphytic crust*" OR "microbiotic crust*" OR "cyanobacterial*" OR "algal*" OR "lichen*" OR "moss*" OR "biotic crust*") AND ("mapping*" OR "distribution*" OR "spatial pattern*") AND ("dryland" OR "hyper*arid*" OR

"arid*" OR "semi*arid*" OR "dry subhumid*"), with research interests in Environmental Sciences/Ecology and a total of 700 papers. (c) Global biocrust data distribution, based on field surveys and literature compilation. Data have been collected and expanded from the published database (Chen et al., 2020; Rodriguez-Caballero et al., 2018) to 3848 items.

*In section 3.3, there should be a connection between the importance of precipitation with the previously mentioned studies of biocrust distribution. This can be done again, by using a map. The authors can make a map, using publicly available data, showing the global precipitation patterns next to the models of the global biocrust distribution. The same can be done with temperature.*

**Response: In the revised manuscript, we have added a new Figure 4 as you suggested**, to show biocrust distribution map with environmental maps (with precipitation and soil texture as examples, Fig. 4), and cited the figure in the corresponding places (lines 244, 279, 361).

[Figure]

Fig. 4 Biocrust distribution and its critical influencing factors. (a) Biocrust cover map and its influencing factors. (a) Global biocrust distribution, by random forest modelling. Based on a global biocrust database constructed by Chen et al., we expanded the biocrust data to 3848 entries through literature compilation and field surveys and fitted them with four types of remotely sensed environmental data, including climate, land use, soil properties, and elevation, to finally predict the suitable areas for the biocrust distribution and quantify the biocrust cover. (b) Global average annual precipitation (1970-2020), data from the WorldClim database (version 2.1). (c) Global soil texture distribution, data from HWSD (Harmonized World Soil Database, version 1.2). Precipitation and soil texture were taken as examples of environmental factors.

*L262: there is an abrupt shift from soil variables to fire which does impact biocrust distribution, but this should be a separate section, perhaps with other anthropogenic impacts on biocrust distribution. I think section 3 would work better if it just included a review of local scale and global scale studies. Then there should be an additional section 4 which reviews the climatic, abiotic (i.e. soil texture), and disturbance (natural and anthropogenic) effects on biocrust distribution.*

**Response: Thanks for pointing out this issue and giving this wonderful suggestion. In the revised manuscript,** we have totally reorganized section 3.3, where this issue is located at, to be a new section 4, to discuss the effects of precipitation, temperature, soil variables, fire, disturbance and other factors on biocrust distribution in separate paragraphs, for a smoother logical flow. (lines 244-320)

*I enjoyed section 4 and like your proposed methods of solving some of the issues with measuring biocrust distribution. This information would also be helpful to have in a table like you did with Table 1, highlighting the advantages and disadvantages. However, in agreement with the other referees, there should be a concluding paragraph summarizing the goals and results of this paper.*

**Response: We are pleased with your endorsement of section 4 of the article.** We would like to agree with you on giving a table to summarize the advantages and disadvantages of the proposed approaches and measure to advance studies on biocrust distribution. Whereas we decided to not add such a table, because the contents and major points of each methods/measures differ dramatically. For example, new section 5.1 (original section 4.1) is about database; new section 5.2 (original section 4.2) discusses modelling approaches; new section 5.3 touches equipment – improving biocrust-specific remote sensing products; section 5.4 goes to algorithms and statistics; section 5.5 talks about social aspect. Therefore, it may be distracting to combine so different aspects in one table. Even this table is given, readers still need to go back to main text to get detailed information as the table is unable to convey so contrasting information. In this sense, this table differs from the present Table 1, which is about available methods to study biocrust distribution. More importantly, section 5 is to inform readers about challenges and ways to go ahead, and it should focus on promising side relative to negative side. We hope you can agree with us.

**In the revised version,** we have added the conclusion section as follows "This work aims to advance global knowledge of biocrust distribution for better ecosystem management and sustainable development in drylands. We firstly compared the advantages, disadvantages, and applicability among three methods, spectral characterization index, dynamic global vegetation models and geospatial models, in order to provide the most appropriate methodological suggestions for biocrust distribution studies at different scales and needs. Then, we systematically sorted out the regional-global biocrust distribution cases, and drew a map of global biocrust distribution hotspots and a map of spatial distribution of data points. Further, we tried to clarify the causes of biocrust distribution from several aspects, such as precipitation, temperature, soil, fire, and other anthropogenic factors. Finally, from a personal point of view, we would like to focus more on the following points in the future: database construction, model performance enhancement, big data processing, and synergistic progress of potential distribution area studies.". (lines 455-466)

*Technical Corrections*

*L49: typo in the citation "(KrÖPfl et al., 2022)"*
**Response: Done.** (line 52)

*L174-178: check the phrasing of this sentence, it should be two separate sentences*
**Response: This sentence has been rewritten as two separate sentences "**In the Mojave Desert, biocrusts distribution was closely related to geological age, surface stability, topography, and dust

transport (Miller et al., 2004)." and "Lichen, moss, and dark algal crusts patchily distributed on the desert, averaging 8% cover, though in some bar and shrub zones, the cover could be as high as 26% (Pietrasiak et al., 2014)." (lines 186-188, 188-190).

*L185, 228: avoid using questions in the text, simply state the results*

**Response: Thanks for your suggestions, and we have changed these two expressions to declarative sentences in the revised version "**In the Loess Plateau, RGB image-based biocrusts monitoring showed that variability in biocrusts cover decreased logarithmically with increasing plot size until a critical size of 1m2 after which biocrusts cover remained approximately constant (Wang et al., 2022a)." (lines 197-200) and "Numerous experimental observations and modelling (Kidron and Xiao, 2023; Li et al., 2023; Rodriguez-Caballero et al., 2018) have proved that, on the global scale, biocrust distribution is mainly influenced by water conditions, temperature, soil properties, fire and disturbance (Bowker et al., 2016)." (lines 240-243).

*L262: there is an abrupt shift from soil variables to fire*

**Response: As above in response to you and reviewer 2,** here we have set up a separate paragraph on the effects of fire.

*L267: Brianne should be Palmer et al. 2020*

**Response: Done (line 292).**

*L274: Please summarize Bowker 2016 then cite correctly*

**Response: Bowker 2016 has been clarified and correctly cited.** "For further insights, readers are encouraged to consult Chapter 10 of Biological Soil Crusts: An Organizing Principle in Drylands, which overview of the control and distribution patterns of biocrust from micro to global scales (Bowker et al., 2016)." (lines 317-320)

*L289: check the grammar*

**Response:** We have checked the grammar of this sentence and rewritten it as "Due to the difficulty of conducting field surveys worldwide, compilating biocrusts data from the published literatures or other sources would be the major approach (Fig. 4(a))." (lines 359-361). In the revised manuscript, we have carefully checked the whole main text by ourselves and by ChatGPT.

*L290-292: rephrase this sentence, the grammar is off*

**Response: The sentence has been rewritten as** "To date, several published studies have assembled 900 ~ 1,000 data on biocrust presence or absence from the literature (including 584 data on biocrust cover)". (lines 361-362)

---

## Referee Report (RR1)

Advancing studies on global biocrust distribution
Wang et al.

**Summary:**

The authors present a thorough review paper about the methods of measuring biocrust distribution, the factors that impact biocrust distribution, and the challenges in mapping biocrust distribution. Where the paper really shines, is the author's straight-forward descriptions of the modeling techniques and methods. This would be helpful to biocrust researchers in the future and is a nice starting point for distribution studies. Overall, I recommend this paper for publication with *minor revisions*.

**Reviewer Comments:**

Overall, there are no major concerns. I know the author's worked very hard editing this draft and it shows. Well done to everyone involved. Below, I have a few line comments to improve the clarity of the manuscript.

Line 21: "still needs to be" should be "remains limited"

Line 22: "stimulate" should be "simulate"

Line 29: "is supposed to" should be "will"

Lines 60-63: This is true, but I don't see why it is necessary in the introduction for this paper. It does not provide any new information about biocrust distribution studies and the papers are cited elsewhere in the introduction.

Figure 1a: I think it is more suitable to show where biocrust distribution is measured (like 1b and 1c) rather than emphasizing the authors because unless the reader knows exactly where those authors do most of their work, it does not provide any new information.

Figure 1: 1a is very difficult to read since it is blury (though see above comment) and the font size of 1b and 1c is also too small. Perhaps you can modify the figures so the continent text size (1b) is larger and the font size of the graph in 1c is larger.

Line 104: "inverted" should be "invented?" I am unsure what the authors are saying here

Figure 2: make the font size a little bigger. This will fill the white space and make it easier to read.

Line 258: "For a long time" is not necessary

Line 268: "The grassland is…" or "Grasslands are…"

Line 273: parentheses error with Condon and Pyke

Line 302: "To sum up" is not necessary

Figure 4: These figures are much easier to read and very nice ☺

Line 326 and 238: Chapter should be Section

Line 365: avoid using "very"

---

## Author Response (AR2)

**Editor's suggestion**

Dear Wang et al,

I have reviewed your revision of the manuscript "Advancing studies on global biocrusts distribution" submitted to SOIL. I appreciate the extensive revisions made to Figure 2 and the addition of Figure 4, both bring important context and synthesis to the work.

However, all three reviewers mentioned the need for the manuscript to be reorganized and include more synthesis and perspective on the work reviewed. The edits on this front have been minimal, consisting primarily of relabeling Section 3.3 as Section 4 and adding a Conclusion paragraph. I believe the reviewers, and myself, intended to recommend extensive reorganization, with text grouped in new logical ways that might improve the reader experience. Perhaps bring stronger emphasis to how results from the various research approaches may differ and how that in turn may influence global biogeographic patterns? Rather than emphasizing the technicalities of the research methods. Reconsidering the way this information builds could help generate more synthesis with clear take-home messages. The revised concluding paragraph primarily outlines the paper structure, without iterating clear synthesis of the literature. What did you learn from reviewing these studies? Do they provide new insights when taken together that none could achieve individually? What are the major knowledge gaps, and how can the field move forward? The final sentence does not provide any insight into actional next steps for this research team or others in the field.

As such, I recommend another revision of the manuscript with greater attention to larger ideas, synthesis, and flow of the information presented.

Elizabeth Bach
* * *
**Response**

Dear Elizabeth,

**Many thanks for your efforts to process this manuscript and valuable suggestions for improving it.** We are pleased to see your positive feedback on Figure 2 and Figure 4 added in the previous version. After carefully checking your suggestions, we have made the following changes:

**Structural reorganization:**
We agree with you that it's essential to emphasize differences in biocrust distribution resulting from various research approaches. To that end, in the revised manuscript, we split the original Chapter 3, "Current State of Knowledge" into two parts:
a) The bibliometric section has been moved to the "Introduction" chapter to link past research hotspots with an overview of our work.
b) The regional and global biocrust distribution sections have been integrated into the current Chapter 2, "Research Methods." In this new chapter, three research approaches were introduced but were not the focus. In contrast, biocrust distribution was given more attention. Meanwhile, we also included more case studies to compare biocrust distribution results under different methods.

Chapter 3, "Influencing Factors of Biocrust Distribution," aims to help researchers and readers fully understand the basis for selecting environmental data sources in biocrust distribution studies. We added transitional segments to this chapter in the revised manuscript.

Chapter 4, "Challenges and Perspectives", aims to expand the topic beyond comparing methods and current understandings, offering a broader perspective to advance studies on biocrust.

**Conclusion Revisions:**
In the new manuscript, the recapitulation of the article structure has been removed. Instead, we have reorganized the discussion around the issue of global biocrust distribution itself, highlighting the current research limitations and the necessity of this paper. The concluding section, based on Chapter 4, "Challenges and Perspectives", now provides a more systematic synthesis and clearer action guidelines.

Once again, thank you for your insightful suggestions. We hope these revisions meet your expectations and stimulate readers' thinking about biocrust distribution prediction.

Best regards,
Siqing Wang

---

## Author Response (AR3)

**Referee 2**

Thank you for the substantial revision that was done; the flow was much smoother and I think the structure helped highlight the key results and novel directions. The major focus for the final revision should be on ensuring that the figures have complete information in the captions and legends.

I will note that it was time consuming to do the second review when there were no line numbers in the response to reviewers indicating where specific comments had been updated. I suggest in future revisions that the authors respond to each comment directly rather than "The response to reviewer X has been uploaded as an attachment."

Response: We are so happy that you are satisfied with our revision and grateful for the key suggestions you have given to improve this article. In the revised manuscript, we have addressed all the points raised by you. Please find the following sections and the main text. Also, thanks for the kind reminder that I will take care to respond directly to reviewers' comments in future revisions.

L 96. I don't see any drylands in the UK or Germany: https://www.fao.org/dryland-assessment/es/

Response: Thank you for pointing this out. For outlining drylands, we used Aridity Index data from Zomer et al. (2022). There is no dryland in the UK, but scarce areas of Germany belong to drylands. To simplify the expression, in the revised manuscript, we have revised this sentence to "Countries like China, the United States, Spain, Australia, and Israel, most of which have extensive dryland areas, have attempted to make breakthroughs on this issue (Fig. 1a)" (lines 59~61). The revised Figure 1(a) is also modified accordingly.

L255. Spell out RGB prior to using abbreviation – since this is for a general audience and not a remote sensing audience this will not necessarily be common knowledge.

Response: The full name of the RGB has been spelled out. (Line 109)

Fig 1. Caption needs substantial additional detail. A. what are the node sizes and distances? b. is this the origin of the author or the location of the study? c. What is an occurrence point? Is it just a publication originating from or highlighting that region? Or is it a dataset?

Response: A. Taken together with reviewer 3's suggestion, the original Figure 1(a) does not provide much valuable information on the topic of biocrust distribution, and the authors' research has been cited elsewhere in the review. In the revised manuscript, the original Figure 1(a) has been removed.
B. An additional description has been added "Numbers are the countries of the authors of published articles from 1990 to 2022⋯" (lines 68~69)
C. Occurrence point refers to the number of record items for which biocrust presence or absence or biocrust cover is noted. It is a dataset (unpublished) that we have continuously updated through field surveys and literature compilation. (lines 74~78)

Fig 4. A. Biocrust cover needs units, or word "proportion" in the caption. B. precipitation needs units.

Response: Done! Thanks!

**Reference**

Robert J. Zomer, Jianchu Xu, Antonio Trabucco. 2022. Version 3 of the Global Aridity Index and

Potential Evapotranspiration Database. **Scientific Data** 9:409.

**Referee 3**

The authors present a thorough review paper about the methods of measuring biocrust distribution, the factors that impact biocrust distribution, and the challenges in mapping biocrust distribution. Where the paper really shines, is the author's straight-forward descriptions of the modeling techniques and methods. This would be helpful to biocrust researchers in the future and is a nice starting point for distribution studies. Overall, I recommend this paper for publication with minor revisions.

Response: Thank you for your recognition and efforts in improving this paper. We hope that the review paper will provide a better understanding of the current state of biocrust distribution research, as well as give researchers specific suggestions on the next practical directions to take.

Reviewer Comments:

Overall, there are no major concerns. I know the author's worked very hard editing this draft and it shows. Well done to everyone involved. Below, I have a few line comments to improve the clarity of the manuscript.

Line 21: "still needs to be" should be "remains limited"

Line 22: "stimulate" should be "simulate"

Line 29: "is supposed to" should be "will"

Response: Done! Thanks! (line 21, 22, 29)

Lines 60-63: This is true, but I don't see why it is necessary in the introduction for this paper. It does not provide any new information about biocrust distribution studies and the papers are cited elsewhere in the introduction.

Response: Thanks for the advice. As the work of these researchers has been referenced elsewhere in the text, we have removed the separate introductions to the authors in the revised manuscript.

Figure 1a: I think it is more suitable to show where biocrust distribution is measured (like 1b and 1c) rather than emphasizing the authors because unless the reader knows exactly where those authors do most of their work, it does not provide any new information.

Response: Thanks for the suggestion. The revised Figure 1 removed the original panel 1a.

Figure 1: 1a is very difficult to read since it is blury (though see above comment) and the font size of 1b and 1c is also too small. Perhaps you can modify the figures so the continent text size (1b) is larger and the font size of the graph in 1c is larger.

Response: The font size has been modified.

Line 104: "inverted" should be "invented?" I am unsure what the authors are saying here

Response: Done! (line 99)

Figure 2: make the font size a little bigger. This will fill the white space and make it easier to read.

Response: Done.

Line 258: "For a long time" is not necessary

Line 268: "The grassland is…" or "Grasslands are…"

Response: Done. (line 253, 263)

---

## Author Response (AR4)

Dear Wang et al.,

Thank you for your continued efforts to finalize this manuscript. I agree with both reviewers that this manuscript is ready for publication, congratulations! I noticed two minor items that should be revised before the final proofs.

Thank you for your hard work revising this manuscript. I really appreciate your responsiveness to the reviewer feedback and my own. The manuscript has matured through the review process. I'm pleased to share this work in SOIL.

Elizabeth Bach,
Topical Editor
* * *
Response: Many thanks for your patience and suggestions. We are so glad that you and two reviewers are satisfied with the manuscript. It is our sincere hope that through continued revision, this work will be seen by a wider audience, as well as provoke SOIL readers to think about the distribution of biological soil crusts.

Line 228: "Permissions have been obtained from the relevant sources." – What are these sources? It's best practice to list them.
Response: In the revised manuscript, this sentence has been rephrased as "Permissions have been obtained from the relevant sources Porada et al., (2019) and Rodriguez-Caballero et al., (2018)." (lines 193~194)

Line 330: "Albert" – is this supposed to be "All be it"? I think this is just an autocorrect typo.
Response: Thanks for pointing out. In the revised manuscript, 'Albert' has been deleted "In addition, factors such as vascular plant cover, topography, and solar radiation also influence biocrust distribution, to a lesser extent than the factors mentioned above." (line 293~295)